**Brief Investigation**

# Revisiting the role of the spindle assembly checkpoint in the formation of gross chromosomal rearrangements in *Saccharomyces cerevisiae*

Yue Yao,[†] Ziqing Yin,[†] Fernando R. Rosas Bringas (ID), Jonathan Boudeman, Daniele Novarina (ID), Michael Chang (ID) *

European Research Institute for the Biology of Ageing, University of Groningen, University Medical Center Groningen, Antonius Deusinglaan 1, 9713 AV Groningen, The Netherlands

*Corresponding author: European Research Institute for the Biology of Ageing, University Medical Center Groningen, Antonius Deusinglaan 1, 9713 AV Groningen, The Netherlands. Email: m.chang@umcg.nl
[†]These authors contributed equally to this work.

Multiple pathways are known to suppress the formation of gross chromosomal rearrangements (GCRs), which can cause human diseases including cancer. In contrast, much less is known about pathways that promote their formation. The spindle assembly checkpoint (SAC), which ensures the proper separation of chromosomes during mitosis, has been reported to promote GCR, possibly by delaying mitosis to allow GCR-inducing DNA repair to occur. Here, we show that this conclusion is the result of an experimental artifact arising from the synthetic lethality caused by the disruption of the SAC and loss of the *CIN8* gene, which is often lost in the genetic assay used to select for GCRs. After correcting for this artifact, we find no role of the SAC in promoting GCR.

Keywords: spindle assembly checkpoint; mitotic checkpoint; chromosomal rearrangement; interstitial telomere sequence; de novo telomere addition; yeast

## Introduction

Gross chromosomal rearrangements (GCRs) are large-scale changes in the structure of chromosomes. GCRs, which include interstitial deletions, duplications, inversions, and translocations, can affect the number, position, and orientation of genes within a chromosome or between chromosomes. They can occur spontaneously during cell division or as a result of exposure to environmental factors such as radiation or chemical mutagens. GCRs are associated with several genetic diseases, are frequently observed in cancer cells, and can contribute to the initiation or progression of cancer (Chen *et al.* 2010; Alonso and Dow 2021).

The mechanisms that suppress the formation of GCRs have been best studied in the budding yeast *Saccharomyces cerevisiae* using genetic assays, such as the "classical" GCR assay developed by Chen and Kolodner, and variations of this assay (Chen and Kolodner 1999; Putnam and Kolodner 2017). In the classical GCR assay, 2 counterselectable markers, *URA3* and *CAN1*, are located on the left arm of chromosome V between the telomere and *PCM1*, the most telomere-proximal essential gene. A GCR involving the loss of both markers renders the cell resistant to 5-fluoroorotic acid (5-FOA) and canavanine. Using these assays, many GCR-suppressing pathways have been identified. These pathways are involved in processes such as DNA replication and repair, S-phase checkpoints, chromatin assembly, telomere maintenance, oxidative stress response, and suppression of R-loop accumulation (Putnam and Kolodner 2017).

Several pathways are also known to promote GCR formation. Among these, de novo telomere addition, nonhomologous end-joining (NHEJ), and homologous recombination (HR) are notably well characterized (Putnam and Kolodner 2017). De novo telomere addition occurs when a broken chromosome end is healed by the addition of a new telomere, resulting in truncation of the chromosome. NHEJ and HR are the two main pathways for the repair of double-strand breaks, but inappropriate NHEJ and HR can lead to translocations or interstitial deletions. However, deletion of genes important for NHEJ and HR often does not reduce, and can even increase, the rate of GCRs, because NHEJ and HR act to both suppress and generate GCRs (Putnam and Kolodner 2017). In addition, transcription can promote GCR, likely due to transcription-dependent replication stress (Sikdar *et al.* 2008). The Rad1–Rad10 endonuclease also promotes GCR, but how it does so remains unclear, with multiple mechanisms proposed (Hwang *et al.* 2005; Sikdar *et al.* 2008). Lastly, the spindle assembly checkpoint (SAC), the Bub2–Bfa1 GTPase-activating protein complex, and the Ctf8-Dcc1-Ctf8-RFC complex have all been implicated in GCR formation induced by various genetic mutations (Myung *et al.* 2004). The SAC ensures accurate chromosome separation during mitosis by delaying the metaphase/anaphase transition until all kinetochores are attached to microtubules (Marston 2014); the Bub2–Bfa1 complex prevents premature mitotic exit (Scarfone and Piatti 2015); and the Ctf18-Dcc1-Ctf8-RFC complex is important for preventing chromosome loss and precocious sister chromatid separation (Arbel *et al.* 2021). It was proposed that DNA lesions that lead to GCR activate the SAC and delay mitotic exit, allowing time for GCR-inducing repair to occur; without this cell cycle delay, cells would progress through mitosis before the damage can be repaired,

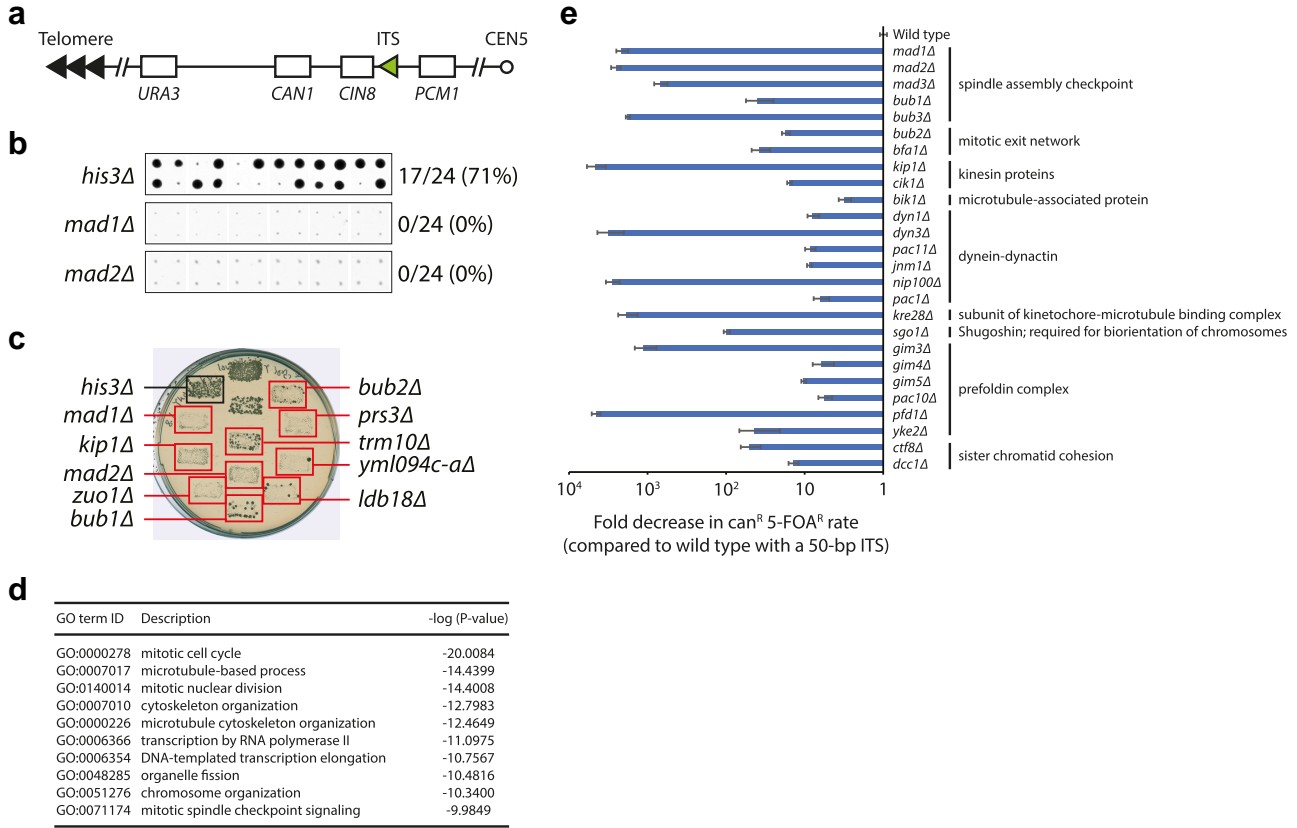

**Fig. 1.** A genome-wide screen for genes that promote the formation of ITS-induced GCRs identifies genes with functions in microtubule-based processes and chromosome segregation. a) Schematic diagram of the ITS-GCR assay. A GCR that leads to the simultaneous loss of two genetic markers, *URA3* and *CAN1*, can be selected by growth on 5-FOA and canavanine. A 50-bp ITS was inserted between *CIN8* and *PCM1*, the most telomere-proximal essential gene on the left arm of chromosome V. b) A high-throughput screen was performed (Rosas Bringas *et al.* 2024). All 24 replica-pinned colonies on media containing both canavanine and 5-FOA of the *his3Δ* control strain and two selected mutants with decreased GCR frequencies are shown. c) Putative hits were tested in a patch-and-replica-plate assay. An example plate is shown. Hits that tested positive are indicated by red boxes. A negative control (*his3Δ*) and a positive control (*bub2Δ*) were included on each plate. d) The top 10 GO terms (Ashburner *et al.* 2000; The Gene Ontology Consortium *et al.* 2023) enriched in the hits that tested positive in the patch-and-replica-plate assay are shown. e) Fold change in canavanine/5-FOA-resistance rate of the indicated strains, relative to the wild-type strain with a 50-bp ITS, is plotted. Error bars represent SEM ($n = 3–6$). The same data normalized to the wild-type strain without a 50-bp ITS are shown in Supplementary Fig. 1.

causing increased lethality and an apparent suppression of GCRs (Myung *et al.* 2004).

To explore the impact of interstitial telomeric sequences (ITSs) on GCR, we modified this assay by inserting a 50-bp ITS between *PCM1* and *CAN1* (Fig. 1a). This modification results in a >1000-fold increase in the GCR rate (Rosas Bringas *et al.* 2024). Subsequently, we performed a genome-wide screen and identified genes that promote ITS-induced GCR, including SAC genes, *BUB2* and *BFA1*, and *CTF8* and *DCC1*, consistent with the previous finding that these genes play a role in GCR formation (Myung *et al.* 2004). However, we find that the apparent GCR-suppressing effect of these mutants can be attributed to the known synthetic lethality arising from the deletion of any of these genes combined with the loss of *CIN8* (Geiser *et al.* 1997; Hardwick *et al.* 1999), which encodes a bipolar kinesin motor protein that plays a pivotal role in mitotic spindle assembly and chromosome segregation (Hoyt *et al.* 1992; Saunders and Hoyt 1992). *CIN8* is located immediately telomere-proximal of the inserted ITS (Fig. 1a), and is often lost during GCR formation in the classical GCR assay. We find that SAC, *bub2Δ*, *bfa1Δ*, *ctf8Δ*, and *dcc1Δ* mutants do not suppress GCRs in strains with an additional copy of *CIN8* located elsewhere in the genome. Therefore, we conclude that the SAC, Bub2–Bfa1, and the Ctf18-Dcc1-Ctf8-RFC complex do not significantly contribute to GCR formation.

## Materials and methods
### Yeast strains and plasmids

All yeast strains used in this study are listed in Supplementary Table 1. Standard yeast genetic and molecular methods were used (Sherman 2002; Amberg *et al.* 2005). Strains containing an extra copy of *CIN8* were constructed as follows: *CIN8* with its own promoter and terminator was PCR-amplified from yeast genomic DNA and cloned via BsaI Golden Gate assembly into plasmid pYTK164 (GFP-dropout *HO*-locus integration vector constructed with the MoClo-YTK) (Lee *et al.* 2015). The resulting plasmid (pDN60.3) was digested with NotI for the integration of *CIN8* at the *ho* locus.

### High-throughput replica-pinning screen

The high-throughput replica-pinning screen was performed as previously described (Novarina *et al.* 2022; Rosas Bringas *et al.* 2024).

### Fluctuation tests of GCR rates

Fluctuation tests for the quantification of GCR rates were performed essentially as previously described (Putnam and Kolodner 2010) by transferring 5 entire single colonies from YPD plates to 4 ml of YPD liquid medium and grown to saturation.

Fifty microliters of a $10^5$-fold dilution was plated on YPD plates. A strain-dependent quantity of cells was plated on SD-arg + canavanine, 5-FOA plates. Colonies were counted after incubation at 30°C for 3–5 days. The number of GCR (can$^R$ 5-FOA$^R$) colonies was used to calculate the GCR rate by the method of the median (Lea and Coulson 1949). Each experiment was performed 3–6 times.

## Gene ontology enrichment analysis

The GO term finder tool (http://go.princeton.edu/) was used to query biological process enrichment for each gene set, with a P-value cutoff of 0.01 and Bonferroni correction applied (Ashburner *et al.* 2000; Boyle *et al.* 2004; The Gene Ontology Consortium *et al.* 2023). REVIGO (Supek *et al.* 2011) was used to further analyze the GO term enrichment data, using the "Medium (0.7)" term similarity filter and the simRel score as semantic similarity measure. As a result, terms with a frequency more than 10% in the REVIGO output were eliminated for being too broad.

## Results and discussion

To identify genes that promote the formation of ITS-induced GCRs, we modified the classical GCR assay (Chen and Kolodner 1999) by inserting a 50 bp ITS between the most telomeric essential gene on the left arm of chromosome V (PCM1) and two counterselectable markers (CAN1 and URA3) (Fig. 1a). We used this ITS-GCR reporter to screen the yeast knockout (YKO) and conditional temperature-sensitive (ts) strain libraries (Rosas Bringas *et al.* 2024). A total of 213 YKO and 93 ts hits were identified in the screen (Fig. 1b) and confirmed in a patch-and-replica-plate assay (Fig. 1c). The hits are enriched for genes involved in the mitotic cell cycle and microtubule-based processes, including the SAC (Fig. 1d).

To further validate the hits, we performed fluctuation tests for a subset of mutants. We find that deletion of genes important for the SAC (MAD1, MAD2, MAD3, BUB1, BUB3), the Bub2–Bfa1 complex, and the Ctf18-Dcc1-Ctf8-RFC complex (CTF8, DCC1) reduces the increased GCR rate caused by the ITS (Fig. 1e; Table 1; Supplementary Fig. 1), reminiscent of the previous finding that deletion of these genes can suppress GCRs in mutants with elevated GCR rates, as assayed using the classical GCR assay (Myung *et al.* 2004). Additionally, we find that deletions of many other genes with microtubule-related functions—including those that encode kinesin and microtubule-associated proteins (KIP1, CIK1, BIK1), dynein-dynactin (DYN1, DYN3, PAC11, JNM1, NIP100, PAC1), proteins involved in the attachment of microtubules to kinetochores (KRE28, SGO1), and subunits of the prefoldin complex (GIM3, GIM4, GIM5, PAC10, PFD1, YKE2), which is important for microtubule biogenesis—also decrease the ITS-induced GCR rate (Fig. 1e; Table 1; Supplementary Fig. 1).

While investigating the mechanism by which these genes could promote GCR, we realized that all the gene deletions shown in Fig. 1e have been reported to be synthetic lethal with co-deletion of the CIN8 gene (Geiser *et al.* 1997; Hardwick *et al.* 1999; Pan *et al.* 2004; Tong *et al.* 2004), which encodes a kinesin motor protein (Hoyt *et al.* 1992; Saunders and Hoyt 1992) that is often lost along with CAN1 and URA3 when selecting for GCRs using the classical GCR assay (Fig. 1a). Thus, the apparent suppression of GCRs by these mutants could be explained by the inability of these mutants to survive a GCR event that also results in the loss of CIN8. To examine the real effect of the SAC, the Bub2–Bfa1 complex, and the Ctf18-Dcc1-Ctf8 complex on GCR formation, we determined the GCR rate of strains, with and without the 50-bp ITS, that have an extra copy of CIN8 integrated at the ho locus on chromosome IV. In this setting, any synthetic lethality caused

**Table 1.** Rates of can$^R$ 5-FOA$^R$ for strains containing a 50-bp ITS.

| Genotype | can$^R$ 5-FOA$^R$ rate |
|---|---|
| Wild type (without ITS) | $4.4 \times 10^{-9}$ |
| Wild type | $7.5 \times 10^{-6}$ |
| *mad1Δ* | $3.5 \times 10^{-9}$ |
| *mad2Δ* | $3.0 \times 10^{-9}$ |
| *mad3Δ* | $1.1 \times 10^{-8}$ |
| *bub1Δ* | $1.9 \times 10^{-7}$ |
| *bub3Δ* | $4.2 \times 10^{-9}$ |
| *bub2Δ* | $4.3 \times 10^{-7}$ |
| *bfa1Δ* | $2.0 \times 10^{-7}$ |
| *kip1Δ* | $1.6 \times 10^{-9}$ |
| *cik1Δ* | $4.8 \times 10^{-7}$ |
| *bik1Δ* | $2.4 \times 10^{-6}$ |
| *dyn1Δ* | $9.5 \times 10^{-7}$ |
| *dyn3Δ* | $2.4 \times 10^{-9}$ |
| *pac11Δ* | $8.8 \times 10^{-7}$ |
| *jnm1Δ* | $8.6 \times 10^{-7}$ |
| *nip100Δ* | $2.7 \times 10^{-9}$ |
| *pac1Δ* | $1.2 \times 10^{-6}$ |
| *kre28Δ* | $4.1 \times 10^{-9}$ |
| *sgo1Δ* | $7.7 \times 10^{-8}$ |
| *gim3Δ* | $6.8 \times 10^{-9}$ |
| *gim4Δ* | $1.2 \times 10^{-6}$ |
| *gim5Δ* | $7.3 \times 10^{-7}$ |
| *pac10Δ* | $1.4 \times 10^{-6}$ |
| *pfd1Δ* | $1.7 \times 10^{-9}$ |
| *yke2Δ* | $1.7 \times 10^{-7}$ |
| *ctf8Δ* | $1.5 \times 10^{-7}$ |
| *dcc1Δ* | $5.4 \times 10^{-7}$ |

**Table 2.** GCR rates of strains with an extra copy of CIN8 at the ho locus, without and with a 50-bp ITS.

| Genotype | No ITS | 50-bp ITS |
|---|---|---|
| Wild type | $1.0 \times 10^{-8}$ | $8.1 \times 10^{-6}$ |
| *mad1Δ* | $1.4 \times 10^{-8}$ | $8.9 \times 10^{-6}$ |
| *mad2Δ* | $2.3 \times 10^{-8}$ | $8.7 \times 10^{-6}$ |
| *mad3Δ* | $2.5 \times 10^{-8}$ | $1.1 \times 10^{-5}$ |
| *bub1Δ* | $3.6 \times 10^{-9}$ | $1.0 \times 10^{-5}$ |
| *bub3Δ* | $5.8 \times 10^{-9}$ | $1.4 \times 10^{-5}$ |
| *bub2Δ* | $8.4 \times 10^{-9}$ | $8.6 \times 10^{-6}$ |
| *bfa1Δ* | $8.8 \times 10^{-9}$ | $5.8 \times 10^{-6}$ |
| *ctf8Δ* | $1.1 \times 10^{-8}$ | $5.8 \times 10^{-6}$ |
| *dcc1Δ* | $2.2 \times 10^{-8}$ | $7.1 \times 10^{-6}$ |

by loss of the endogenous CIN8 gene is circumvented by the presence of the second copy of CIN8. The extra CIN8 does not significantly affect the GCR rate of wild-type strains without an ITS ($4.4 \times 10^{-9}$ vs $1.0 \times 10^{-8}$, without and with the extra copy of CIN8, respectively) or with a 50-bp ITS ($7.5 \times 10^{-6}$ vs $8.1 \times 10^{-6}$, without and with the extra copy of CIN8, respectively; Tables 1 and 2). We then tested the deletion of genes involved in the SAC and the Bub2–Bfa1 and Ctf18-Dcc1-Ctf8 complexes and find only mild (<3-fold) changes in GCR rate (Fig. 2; Table 2). Thus, the SAC, along with the Bub2–Bfa1 and Ctf18-Dcc1-Ctf8 complexes, neither promote nor suppress the formation of GCRs.

Our findings highlight an important point to consider when using genetic assays. If the assay results in the loss of genes not directly related to the assay, it is important to consider whether a synthetic lethal genetic interaction exists between one of these genes and any mutant being studied using the assay. If such a synthetic lethal interaction exists, it (1) may give the false appearance that the mutant decreases the rate of the studied genetic event, and (2) may mask an actual increase in the rate. For the classical GCR assay, there are 23 open reading frames (several of which are classified as dubious in the *Saccharomyces* Genome Database) between

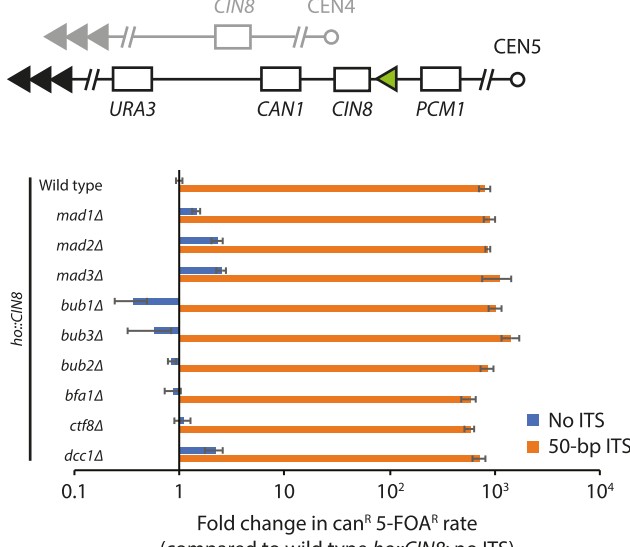

**Fig. 2.** Defects in the SAC, the Bub2–Bfa1 complex, or sister chromatid cohesion cannot suppress ITS-induced GCR rate when an extra copy of *CIN8* is present. Fold change in the GCR rate of the indicated strains, which all have an extra copy of *CIN8* inserted at the *ho* locus, is plotted. Fold changes are relative to the wild-type haploid strain with the extra copy of *CIN8* gene, but without an ITS. Error bars represent SEM ($n = 3$–6).

*PCM1* and the telomere on the left arm of chromosome V (Wong et al. 2023). While loss of any of these genes may pose such a problem, most of the known genetic interactions for this group of genes involves *CIN8*. Other assays involving other regions of the genome will be affected by a different set of synthetic lethal interactions.

## Data availability

Strains and plasmids are available upon request. The authors affirm that all data necessary for confirming the conclusions of the article are present within the article, figures, and tables.
Supplemental material available at GENETICS online.

## Acknowledgments

We thank A. Milias-Argeitis for providing plasmid pYTK164.

## Funding

Y.Y. was supported by an Abel Tasman Talent Program scholarship from the University of Groningen. Z.Y. was supported by a scholarship from the Nanjing Huimou Medi-Tech Co. F.R.R.B. was supported by a Consejo Nacional de Ciencia y Tecnología (CONACYT) scholarship. Work in the laboratory of M.C. was supported by an Open Competition M-2 grant from the Dutch Research Council.

## Conflicts of interest

The author(s) declare no conflict of interest.

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

*Editor: D. Bishop*