## [Peer Review File · Genetics]

Revisiting the role of the spindle assembly checkpoint in the formation of gross chromosomal rearrangements in *Saccharomyces cerevisiae*

Yue Yao, Ziqing Yin, Fernando Rosas Bringas, Jonathan Boudeman, Daniele Novarina, and Michael Chang

NOTE: The reviews and decision letters are unedited and appear as submitted by the reviewers.

In extremely rare instances and as determined by a Senior Editor or the EIC, portions of a review may be redacted. If a review is signed, the reviewer has agreed to no longer remain anonymous.

The review history appears in chronological order.

Review Timeline:

Submission Date:	2024-07-01
Editorial Decision:	2024-07-29
Resubmission Received:	2024-09-03
Accepted:	2024-09-04

July 29, 2024

GENETICS-2024-307241

Revisiting the role of the spindle assembly checkpoint in the formation of gross chromosomal rearrangements in *Saccharomyces cerevisiae*

Dear Dr. Chang:

Two experts in the field have reviewed your manuscript, and I have read it as well. I am pleased to inform you that, with minor revisions, it is potentially suitable for publication in GENETICS. The first reviewer has comments and concerns that need to be addressed in a revised manuscript. You can read the two reviews at the end of this email.

I think all of reviewer 1's comments are worth addressing and I expect you will not find it particularly difficult to do so.

We look forward to receiving your revised manuscript. Please let the editorial office know approximately how long you expect to need for revisions.

Upon resubmission, please include:

1. A clean version of your manuscript;
2. A marked version of your manuscript in which you highlight significant revisions carried out in response to the major points raised by the editor/reviewers (track changes is acceptable if preferred);
3. A detailed response to the editor's/reviewers' comments and to the concerns listed above. Please reference line numbers in this response to aid the editors.

Additionally, please ensure that your resubmission is formatted for GENETICS.

<https://academic.oup.com/genetics/pages/general-instructions>

Follow this link to submit the revised manuscript: Link Not Available

Sincerely,

Douglas Bishop
Associate Editor
GENETICS

Approved by:
Amy MacQueen
Senior Editor
GENETICS

Reviewer #1 (Comments for the Authors (Required)):

This manuscript explores a caveat of the classic Kolodner GCR assay in that all clones recovered must necessarily delete a short segment of *S. cerevisiae* chromosome V left arm, with all genes present in it. While the few genes distal to PCM1 are not essential for viability as single deletions, some of them may have synthetic interactions with other genes, such as is the case of CIN8 with other important genes in the spindle assembly checkpoint pathway as demonstrated in this study. This and other caveats of the GCR assay are rarely acknowledged, so this manuscript would be a valuable contribution to correct the literature and also to remind the community of the limitations of the GCR assay. Nevertheless, I did have a few important concerns about the way the data were presented in the manuscript, and also regarding some of the experimental design (particularly the GCR assays done in diploids, which in my opinion are ambiguous and should be removed).

Below are major and minor comments and suggestions that authors and editors should consider to improve the manuscript:

Line 93. Replace "downstream" with "telomere-proximal"

Line 116 and 120. I found the description of the GCR rate measurement experimental methods to be inadequate. For example, the reference to "performed essentially as previously described (Luria and Delbruck 1943)" is inaccurate - The general approach (fluctuation) may be similar to L&D, but the actual work is certainly not (bacteria/phage vs. yeast). Also, the statement about the calculation of rates using the "method of the media" is inaccurate (Putnam and Kolodner 2010 did not describe that method for calculating mutation rates: Lea and Coulson did).

Line 142 and Figure 1E. I think in this figure the data should be normalized to the WT experimental strain containing the ITS, instead of the WT without the ITS (currently the 1x reference). The GCR rate in WT without ITS is not as relevant for this work, so should not be used as the 1x reference.

Also, when plotting mutation rates as relative fold change (as is the case in Fig. 1E and elsewhere in the manuscript), it is essential to also include in the main text and in the figure legend the absolute rate measurement for the reference rate strain (for example: WT with ITS is the 1x normalization reference and had an absolute rate of 2.3×10^{-7} GCR events/cell/division). 2.3×10^{-7} is just an example - replace with the actual number.

Line 169 and Figure 2A. The GCRs experiments done in diploids cells (hemizygous for the URA3-CAN1 GCR markers) can produce 5-FOAR + CanR clones through two fundamentally different rearrangement mechanisms: True terminal deletions (as in conventional haploid GCR) and allelic interhomolog mitotic recombination leading to copy neutral loss of heterozygosity. Given that these two very different structural mutation classes occur at vastly different absolute rates (ie. LOH can be far more frequent), the data in Fig 2A cannot be directly interpreted as derived from classic GCRs. While the Hph marker centromeric to the ITS can serve as an indirect indication of the mechanism, this distinction can only be made definitively if the mutational spectrum of GCRs is determined through genomic analyses such as PFGE and CNV (which was not done here).

Figure 2B. The issue with diploids described above might also be associated with the strange result described in Fig 2B where BUB1 is claimed to be haploinsufficient. This may not be the case, and can only be determined with proper genomic analyses of mutational class in multiple independent 5-FOAR CanR clones.

In general, I feel the diploid experiments are not effective nor required to support the main argument of the manuscript, so the diploid data might as well be removed.

On the other hand, the experiments and data shown in Figure 2C, are much more compelling. Yet, those stronger results were barely described in the Results section (just 2 sentences in lines 200 to 204). In my view this is the primary dataset that supports the main finding presented in the manuscript, so they should be emphasized.

However, I still do have a comment of experimental design for this assay as well. A simpler, more direct, and unambiguous way to demonstrate the role that CIN8 plays in these processes would be to build a new haploid GCR strain where the ITS is integrated on the telomeric of CIN8 (between CAN1 and CIN8). Since most GCR clones recovered in this study are due to rearrangements at the ITS, moving the ITS just a few Kb to the telomeric side of the CIN8 should mimic the results obtained with the strain that contains a second copy of CIN8 on chromosome IV, but without the interpretation caveat of CIN8 overexpression.

Reviewer #3 (Comments for the Authors (Required)):

The mechanisms that suppress the accumulation of gross chromosomal rearrangements (GCRs) have been identified in yeast using a simple genetic assay developed by the Kolodner lab. This assay relies on loss of two counter-selectable genes (CAN1 and URA3) that are telomeric to PCM1, the most telomere proximal essential gene on the left arm of chromosome V. Previously, Myung et al. (2004) reported that loss of the spindle assembly checkpoint (SAC), Bub2-Bfa1 complex or the Ctf18-Dcc1-Ctf8 complex, suppresses the hyper-GCR phenotype of mutations representing several pathways, suggesting that they promote GCR formation. In this study, the authors inserted a 50-bp long interstitial telomere sequence (ITS) between PCM1 and CAN1 to identify genes that suppress ITS induced GCRs. They show that the ITS induces the frequency of GCRs by ~1000-fold and this increase is dependent on the SAC and Bub2-Bfa1, and partially dependent on the Ctf18 complex. However, recognizing that these functions are all required for viability of cells lacking CIN8, a non-essential gene located telomeric to the ITS, they tested whether an additional copy of CIN8 could bypass the phenotype caused by loss of the SAC, Bub2-Bfa1 or the Ctf18 complex. Their data clearly show that the SAC, Bub2-Bfa1 and the Ctf18 complex are not required to promote GCRs if the cells retain a functional copy of CIN8. This finding highlights a limitation of assays that select for loss of a "non-essential" region of the chromosome if that region is essential in certain genetic backgrounds.

Associate Editor Comments:

Point-by-Point Response to the Reviewers' Comments:

We are delighted to hear that our manuscript was well received by the editor and reviewers, and we thank them for their constructive comments. Below, you will find our response to each comment. Our responses are in red and preceded by a bullet point.

Reviewer Comments:

Reviewer #1 (Comments for the Authors (Required)):

*This manuscript explores a caveat of the classic Kolodner GCR assay in that all clones recovered must necessarily delete a short segment of *S. cerevisiae* chromosome V left arm, with all genes present in it. While the few genes distal to PCM1 are not essential for viability as single deletions, some of them may have synthetic interactions with other genes, such as is the case of CIN8 with other important genes in the spindle assembly checkpoint pathway as demonstrated in this study. This and other caveats of the GCR assay are rarely acknowledged, so this manuscript would be a valuable contribution to correct the literature and also to remind the community of the limitations of the GCR assay. Nevertheless, I did have a few important concerns about the way the data were presented in the manuscript, and also regarding some of the experimental design (particularly the GCR assays done in diploids, which in my opinion are ambiguous and should be removed). Below are major and minor comments and suggestions that authors and editors should consider to improve the manuscript:*

Line 93. Replace "downstream" with "telomere-proximal"

- We have replaced "downstream" with "telomere-proximal".

Line 116 and 120. I found the description of the GCR rate measurement experimental methods to be inadequate. For example, the reference to "performed essentially as previously described (Luria and Delbruck 1943)" is inaccurate - The general approach (fluctuation) may be similar to L&D, but the actual work is certainly not (bacteria/phage vs. yeast). Also, the statement about the calculation of rates using the "method of the media" is inaccurate (Putnam and Kolodner 2010 did not describe that method for calculating mutation rates: Lea and Coulson did).

- We have updated the references and added text to improve the description of the fluctuation tests.

Line 142 and Figure 1E. I think in this figure the data should be normalized to the WT experimental strain containing the ITS, instead of the WT without the ITS (currently the 1x reference). The GCR rate in WT without ITS is not as relevant for this work, so should not be used as the 1x reference.

- We have changed the figure as suggested. Since our related manuscript (Rosas Bringas et al.) shows similar data that is normalized to WT without the ITS, we have kept the original figure, but moved it to the Supplementary Material.

Also, when plotting mutation rates as relative fold change (as is the case in Fig. 1E and elsewhere in the manuscript), it is essential to also include in the main text and in the figure legend the absolute rate measurement for the reference rate strain (for example: WT with ITS is the 1x normalization reference and had an absolute rate of 2.3×10^{-7} GCR events/cell/division). 2.3×10^{-7} is just an example - replace with the actual number.

- We have added two new tables containing the absolute rate measurements for all strains indicated in Figure 1e and Figure 2.

Line 169 and Figure 2A. The GCRs experiments done in diploids cells (hemizygous for the URA3-CAN1 GCR markers) can produce 5-FOAR + CanR clones through two fundamentally different rearrangement mechanisms: True terminal deletions (as in conventional haploid GCR) and allelic interhomolog mitotic recombination leading to copy neutral loss of heterozygosity. Given that these two very different structural mutation classes occur at vastly different absolute rates (ie. LOH can be far more frequent), the data in Fig 2A cannot be directly interpreted as derived from classic GCRs. While the Hph marker centromeric to the ITS can serve as an indirect indication of the mechanism, this distinction can only be made definitively if the mutational spectrum of GCRs is determined through genomic analyses such as PFGE and CNV (which was not done here).

Figure 2B. The issue with diploids described above might also be associated with the strange result described in Fig 2B where BUB1 is claimed to be haploinsufficient. This may not be the case, and can only be determined with proper genomic analyses of mutational class in multiple independent 5-FOAR CanR clones.

In general, I feel the diploid experiments are not effective nor required to support the main argument of the manuscript, so the diploid data might as well be removed.

- As suggested, we have removed the diploid data.

On the other hand, the experiments and data shown in Figure 2C, are much more compelling. Yet, those stronger results were barely described in the Results section (just 2 sentences in lines 200 to 204). In my view this is the primary dataset that supports the main finding presented in the manuscript, so they should be emphasized.

However, I still do have a comment of experimental design for this assay as well. A simpler, more direct, and unambiguous way to demonstrate the role that CIN8 plays in these processes would be to build a new haploid GCR strain where the ITS is integrated on the telomeric of CIN8 (between CAN1 and CIN8). Since most GCR clones recovered in this study are due to rearrangements at the ITS, moving the ITS just a few Kb to the telomeric side of the CIN8 should mimic the results obtained with the strain that contains a second copy of CIN8 on chromosome IV, but without the interpretation caveat of CIN8 overexpression.

- While we agree that this experiment would further support our conclusion, we feel it is essentially redundant to the experiment we show in Figure 2, especially since the overexpression of CIN8 is rather mild (the second copy is expressed from its native promoter).

Reviewer #3 (Comments for the Authors (Required)):

The mechanisms that suppress the accumulation of gross chromosomal rearrangements (GCRs) have been identified in yeast using a simple genetic assay developed by the Kolodner lab. This assay relies on loss of two counter-selectable genes (CAN1 and URA3) that are telomeric to PCM1, the most telomere proximal essential gene on the left arm of chromosome V. Previously, Myung et al. (2004) reported that loss of the spindle assembly checkpoint (SAC), Bub2-Bfa1 complex or the Ctf18-Dcc1-Ctf8 complex, suppresses the hyper-GCR phenotype of mutations representing several pathways, suggesting that they promote GCR formation. In this study, the authors inserted a 50-bp long interstitial telomere sequence (ITS) between PCM1 and CAN1 to identify genes that suppress ITS induced GCRs. They show that the ITS induces the frequency of GCRs by ~1000-fold and this increase is dependent on the SAC and Bub2-Bfa1, and partially dependent on the Ctf18 complex. However, recognizing that these functions are all required for viability of cells lacking CIN8, a non-essential gene located telomeric to the ITS, they tested whether an additional copy of CIN8 could bypass the phenotype caused by loss of the SAC, Bub2-Bfa1 or the Ctf18 complex. Their data clearly show that the SAC, Bub2-Bfa1 and the Ctf18 complex are not required to promote GCRs if the cells retain a functional copy of CIN8. This finding highlights a limitation of assays that select for loss of a "non-essential" region of the chromosome if that region is essential in certain genetic backgrounds.

September 4, 2024

RE: GENETICS-2024-307422

Dr. Michael Chang
University Medical Center Groningen
European Research Institute for the Biology of Ageing
A. Deusinglaan 1
Groningen, N/A 9713 AV
Netherlands

Dear Dr. Chang:

Congratulations, your manuscript entitled "Revisiting the role of the spindle assembly checkpoint in the formation of gross chromosomal rearrangements in *Saccharomyces cerevisiae*" is accepted for publication in GENETICS! Many thanks for submitting your research to the journal.

In preparing your final manuscript (steps outlined below), we encourage you to consider one modification in particular: Please include the citations for GO term analysis (Figure 1d and Methods) according to the guidelines provided here: <https://geneontology.org/docs/go-citation-policy/>.

To Proceed to Publication:

1. Format your article according to GENETICS style: <https://academic.oup.com/genetics/pages/general-instructions>

2. Ensure that you comply with data and community resource citation guidelines:
<https://academic.oup.com/genetics/pages/general-instructions#Data-Policy>

3. Upload your final files at <https://genetics.msubmit.net>

4. Add oupsupport@scipris.com and genetics.oup@novatechset.com (or the domains @scipris.com and @novatechset.com) to your email program's "safe senders" list. You will be contacted by both at various points during the production process.

Notes:

- Your currently-accepted manuscript (unedited, as submitted, reviewed, and accepted) will be published at GENETICS and deposited into PubMed as an Advance Access article. Notify sourcefiles@thegsajournals.org before signing your license if you do not wish to publish your article via Advance Access.

- We invite you to submit an original color figure related to your paper for consideration as cover art. Please email your submission to the editorial office or upload it with your final files. You can submit a small-sized image for evaluation, and if selected, the final image must be a TIFF file 2513px wide by 3263px high (8.375 by 10.875 inches; resolution of 600ppi). Please avoid graphs and small type.

- After files are sent to Oxford University Press we use SciPris to manage article licensing and payment. If you do not have a SciPris account, you will receive an email from no-reply@scipris.com to sign up to use Oxford University Press' author portal. After logging in, follow the online instructions to sign your license and arrange any payment due.

If you have any questions or encounter any problems while uploading your accepted manuscript files, please email the editorial office at sourcefiles@thegsajournals.org.

Sincerely,

Douglas Bishop
Associate Editor
GENETICS

Approved by:
Amy MacQueen
Senior Editor
GENETICS

Review comments (if applicable):